

# Bacterioplankton dark CO₂ fixation in oligotrophic waters

Afrah Alothman[1*], Daffne López-Sandoval[1,2], Carlos M. Duarte[1], Susana Agustí[1]

[1]Red Sea Research Centre, Biological and Environmental Science and Engineering Division, King Abdullah University of Science and Technology (KAUST), Thuwal, 23955, Saudi Arabia

[2]Coastal and Marine Resources Core Lab (CMR), King Abdullah University of Science and Technology (KAUST), Thuwal, 23955, Saudi Arabia

*Corresponding author*: afrah.alothman@kaust.edu.sa

**Abstract**

Dark CO₂ fixation by bacteria is believed to be particularly important in oligotrophic ecosystems. However, only a few studies have characterized the role of bacterial dissolved inorganic carbon (DIC) fixation in global carbon dynamics. Therefore, this study quantified the primary production (PP), total bacteria dark CO₂ fixation (TB$_{DIC}$ fixation), and heterotrophic bacterial production (HBP) in the warm and oligotrophic Red Sea using stable isotope labeling and cavity ring-down spectroscopy ($^{13}$C-CRDS). Additionally, we assessed the contribution of bacterial

DIC fixation (TB$_{DIC}$ %) relative to the total DIC fixation (Total$_{DIC}$ fixation). Our study demonstrated that TB$_{DIC}$ fixation increased the Total$_{DIC}$ fixation from 2.03 to 60.45 µg C L$^{-1}$ d$^{-1}$ within the photic zone, contributing 13.18 % to 71.68 % with an average value of 33.95 ± 0.02 % of the photic layer Total$_{DIC}$ fixation. The highest TB$_{DIC}$ fixation values were measured at the surface and deep (400 m) water with an average value of 5.23 ± 0.45 µg C L$^{-1}$ d$^{-1}$, and 4.95 ± 1.33 µg C L$^{-1}$ d$^{-1}$, respectively. These findings suggest that the non-photosynthetic processes such as

anaplerotic DIC reactions and chemo-autotrophic CO₂ fixation extended to the entire oxygenated water column. On the other hand, the % of TB$_{DIC}$ contribution to Total$_{DIC}$ fixation increased as primary production decreased (R² = 0.45, p <0.0001), suggesting the relevance of increased dark DIC fixation when photosynthetic production was low or absent, as observed in other systems. Therefore, when estimating the total carbon dioxide production in the ocean, dark DIC fixation must also be accounted as a crucial component of the carbon dioxide flux in addition to

photosynthesis.

## 1   Introduction

Bacteria are the central nodes of the microbial loop and play an essential role in the flux of organic carbon in marine

ecosystems through different metabolic pathways (Azam et al., 1983). Most studies on the metabolism of marine bacteria have focused on quantifying the uptake of organic compounds by heterotrophic bacteria and how it relates to bacterial growth and reproduction (Ducklow and Kirchman, 2000; Kirchman, 2000). However, heterotrophic marine bacteria can also metabolize CO₂ through anaplerotic carboxylation reactions, which form the basis of several metabolic pathways (Dijkhuizen and Harder, 1984). Such reactions are essential components of metabolic

pathways in bacteria that enable the synthesis of fatty acids, amino acids, vitamins, and nucleotides (Dijkhuizen and Harder, 1984; Erb, 2011), which also fuel microbial food webs.

Wood and Werkman (1936 first proposed that heterotrophic bacteria contribute to dark dissolved inorganic carbon (DIC) fixation, a discovery that was widely embraced by the scientific community. A decade later, when radioactive



isotope techniques emerged in the field, Steemann-Nielsen (Steemann-Nielsen, 1952) first reported on the possible

importance of dark DIC fixation to the total carbon flux in the ocean, suggesting that it could represent between 1 % and 30 % of photosynthetic $CO_2$ fixation (Steemann-Nielsen, 1952; Nielsen, 1960). Subsequent quantifications in the northern to southern Pacific and Atlantic oceans reported that dark DIC fixation accounted for approximately >10 % of photosynthetic $CO_2$ fixation in temperate and equatorial areas, and between 10 % to 50 % of the light fixation rate in subtropical gyres (Prakash et al., 1991). Indeed, dark $CO_2$ assimilation contributes considerably to

DIC fixation in marine surface water, which has been directly associated with high bacterial activity (Prakash et al., 1991, Li et al., 1993, Markager, 1998, Li and Dickie, 1991). A study that evaluated the role of Arctic bacterial dark $CO_2$ incorporation suggested that the depletion and limitation of labile organic carbon compounds could enhance the utilization of bicarbonate by chemoautotroph or heterotroph microorganisms to achieve metabolic balance (Alonso-Sáez et al., 2010). Additionally, it was assumed that dark $CO_2$ utilization was mainly attributable to bacterial

metabolism, as it was highly positively correlated with heterotrophic bacterial production (Alonso-Sáez et al., 2010). In general, anaplerotic $CO_2$ fixed by bacteria has been estimated to contribute up to 8 % of heterotroph carbon biomass production in the ocean (Romanenko, 1964) and contributes significantly to the carbon flux dynamics of many marine ecosystems (Alonso-Sáez et al., 2010, Yakimov et al., 2014, Zhou et al., 2017., Signori et al., 2017, and Lliro's et al., 2011).


Dark $CO_2$ fixation by marine bacteria is thought to play an essential role under oligotrophic conditions, contributing up to 30 % of bacterial production (González et al., 2008; Palovaara et al., 2014). Genetic studies have demonstrated that increases in the abundance of associated anaplerotic enzyme transcripts coincided with a sudden increase in bacterial metabolism, which could contribute significantly to the total DIC fixation rates in oligotrophic

environments (Baltar et al., 2016). Additionally, the recent discovery of light-driven $CO_2$ incorporation by proteorhodopsin-containing flavobacterium *Polaribacter* sp. highlighted the significant role of anaplerotic metabolism in heterotrophs (González et al., 2008). Therefore, whereas the total primary production of oceanic ecosystems is typically attributed to photosynthesis, dark chemo-autotrophic and anaplerotic metabolism are also important contributors (Baltar and Herndl, 2019).


Dark $CO_2$ fixation by bacteria is, therefore, likely to be highly relevant in the Red Sea, a predominantly oligotrophic, landlocked system with no river inflow and with limited connection to the Indian Ocean (Edwards, 1987; Grasshoff, 1969). These features result in a limited nutrient input and an oligotrophication gradient from south to north (Wafar et al., 2016). The Red Sea is also characterized by high surface temperatures ranging from 20 to 33.1 °C (Chaidez et

al., 2017, Shaltout, 2019) and warm deep-water temperatures of up to ~21.5 °C at depths below 300 m (Yao & Hoteit, 2018).

Here we assess the contribution of dark $CO_2$ assimilation to the Red Sea bacterioplankton production using $^{13}C$ stable isotope as a tracer. In this study, both the dark bicarbonate synthesis by bacteria ($TB_{DIC}$) and light bicarbonate synthesis by photosynthetic phytoplankton (PP) were quantified using $^{13}C$ as a tracer in the Red Sea water column.

Additionally, $^{13}C$ stable isotope fluxes were used to estimate bacterial production (BP) in the dark. Our study also





assessed the variations in dark $CO_2$ fixation rates at different depths through the water column in both open and coastal water bodies, and the relationship with water temperature. Moreover, our study estimated the contribution of dark bicarbonate synthesis ($TB_{DIC}$ %) to total $CO_2$ fixation ($Total_{DIC}$ fixation) by accounting for dark and light $CO_2$ fixation rates.


## 2 Methods

### 2.1 Sample collection and processing

A total of 59 water samples were collected over the course of four oceanographic cruises in the Eastern Red Sea, including the Center Competitive Found (CCF), Deep Cruise (DC), Deep Coral Survey (DCS), and Red Sea Decade Expedition (RSDE). Samples were also obtained from a time series fixed station (pelagic) and two other coastal sites (a lagoon and reef located in Abu Shusha) in the Central Red Sea of Saudi Arabia (Fig. 1). The cruises were conducted on board R/V *Thuwal* (DC, DCS), R/V *Al Azizi* (CCF), and *OceanXplorer* (RSDE) between August 2017

and June 2022. During the CCF oceanographic cruise, water samples were collected at five different depths ranging from the surface to the bottom layer of the photic zone. During the DC and DCS oceanographic cruises, time series and coastal station water samples we collected from surface water at a 3–5 m depth. During the RSDE cruise, water samples were collected from the surface at a 5 m depth, as well as from 400 m. Vertical environmental profiles of temperature, salinity, and photosynthetic active radiation were obtained for all of the studied stations as described by

López-Sandoval et al. (2021).

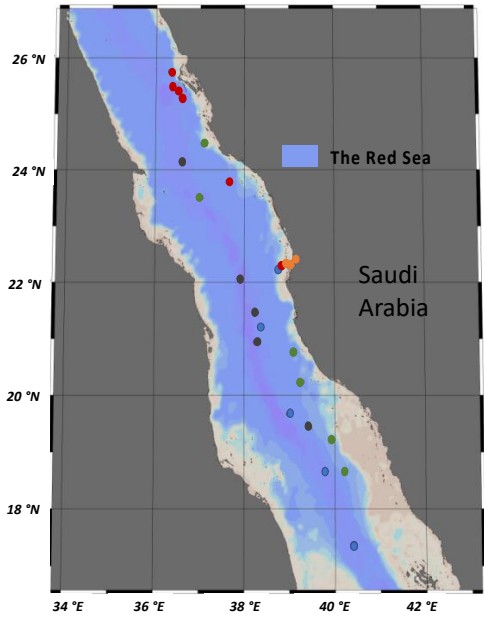



**Figure 1**: Stations sampled during the four oceanographic cruises [CCF (blue dots), DC (green dots), DCS (red dots), RSDE (black dots)] time series and coastal stations (orange dots) along the Eastern Red Sea conducted between 2017 and 2022.

At each station, water samples for isotope labeling analysis were collected in the morning using 12 L Niskin bottles with a rosette system or 10 L Niskin bottles deployed manually (López -Sandoval et al., 2021). For the deepest water (400 m), the water samples were collected using 1.5 L Niskin bottles attached to a remotely-operated underwater vehicle (ROV) and a submarine on board *OceanXplorer*. Water samples were collected directly from the Niskin bottles, prefiltered through 100μm mesh filters to remove larger zooplankton, and transferred into 10 L acid-washed carboy containers. The water samples were distributed into three transparent 2 L or 500 mL ($^{13}$C-PP; López-Sandoval, et al. 2019) polycarbonate (PC) bottles for light primary production measurements and another three dark 2 L or 500 mL PC bottles for measuring dark bacterial DIC uptake rate ($^{13}$C-TB$_{DIC}$). All water samples were enriched with $^{13}$C-sodium bicarbonate solution (99.8 atom % 4 g/L of NaH$^{13}$CO$_3^-$; López-Sandoval et al., 2019) to a final carbon concentration of ~153 μmole $^{13}$C L$^{-1}$ in each bottle. Additionally, during the DC and DCS oceanographic cruises, three dark PC bottles were enriched with $^{13}$C-D-glucose substrate at a final concentration of 100 nM to measure bacterial production ($^{13}$C-BP) as described by Koshikawa et al. (1999). All PC bottles were incubated in tanks placed on the vessel's deck with a circulating seawater system to maintain surface water temperature and receive natural solar radiation. Moreover, a separate tank was attached to a chiller to mimic the water temperature at a 400 m depth. Additionally, coastal water samples were incubated in a similar outdoor setup at the Coastal & Marine Resources Core Lab (CMR) at King Abdullah University of Science and Technology (KAUST).

The bottles for PP were covered with neutral-density nets to reduce the light intensity according to the matching light received at the assigned depth. After 4–6 hours of incubation, the water samples collected during the CCF cruise and the time series station were filtered through pre-combusted Whatman GF/F filters (López-Sandoval et al., 2021). The rest of the samples were filtered through 25 mm diameter 3 μm Silver membranes (STERLITECH) to collect the size fraction above the picoplankton size. Afterward, the filtrate was collected into a 25 mm diameter (0.2 μm) Silver membrane filter (STERLIECH). The samples collected during the RSDE cruise were pre-filtered through 3 μm polycarbonate membrane filters before incubation. The natural isotopic composition of particulate organic carbon was measured in surface and deep seawater at each station. The collected filters were placed in small Petri dishes containing 150 μl (50 %) HCl to remove carbonate from the filters, allowed to dry for 12 hours, and stored at –20 °C until required for downstream analyses.

### 2.1.1 Chlorophyll-*a* concentration and nutrients

Samples for chlorophyll-a (Chl-a) and nutrient analyses were collected at each depth within the photic layer during the CCF cruise and from surface water during the time series study (López-Sandoval et al., 2021). Samples for Chl-a





analysis were collected from all cruises and coastal stations. The water samples (200–500 ml) were filtered through 25 mm Whatman GF/F filters (0.7 µm) and then extracted in 90 % acetone in the dark as described by Prabowo and Agusti (2019) and López-Sandoval et al. (2021). After 24 hours, the extracted pigment was measured using a Trilogy Fluorometer equipped with a CHL-NA module (Turner Designs; San Jose, USA) calibrated with pure Chl-a (Prabowo and Agusti, 2019). Water samples for inorganic nutrient concentration were collected and frozen until analyzed in the laboratory. Nutrient concentrations were determined with a Segmented Flow Analyzer (SEAL AA3 Analytical Inc.; WI, USA) following standard autoanalyzer methods (Hansen and Koroleff, 1999).

### 2.1.2    Heterotrophic bacteria abundance

The abundance of heterotrophic bacteria was quantified in each water sample. Briefly, 1.8 mL aliquots were obtained from each sample, fixed with 25 % glutaraldehyde, flash-frozen in liquid nitrogen, and stored at –80 °C for later analysis. The samples were stained with SYBR Green I intermediate solution (1:100) for the determination of bacteria cell abundance by flow cytometry (Gasol and Moran, 2016) using either a FACSCanto II (Becton Dickinson) or a Flow Cytometer Cube 8 (Sysmex).

### 2.1.3    Dissolved inorganic carbon (DIC)

The $\delta^{13}C$ of DIC in natural waters and after enrichment with $NaH^{13}CO_3^-$ was analyzed in seawater samples placed in 15 mL small glass tubes and treated with 0.05 % mercuric chloride ($HgCl_2$-Sigma-Aldrich) to stop any biological activity after sampling (Dickson et al., 2007). After fixation, all samples were kept in a dark and cool place until analyzed in the laboratory. DIC measurements were conducted using an AutoMate Prep Device coupled with Picarro's LIAISON interface and IsoCO2 WS-CRDS system (Santa Clara, California. USA).

### 2.1.4    Primary production (PP) and total dark bacteria DIC fixation (TB$_{DIC}$)

The carbon content and the $\delta^{13}C$ values from PP and TB$_{DIC}$ incubation filters were analyzed using a combustion module (CM) attached to a cavity-ring down spectroscopy analyzer (CM-CRDS-G2201-I, Picarro), and each filter was analyzed for 600 s. The combustion module converted the sample into the required gas ($CO_2$) by fast combustion, after which the gas was transferred to the isotopic analyzer to measure the $^{13}C/^{12}C$ ratio ($\delta^{13}C$). Inside the cavity, spectrum peaks were generated according to the measured wavelength absorbed by the gas of interest ($^{13}CO_2$ and $^{12}CO_2$), where each peak corresponded to the $^{13}C$ and $^{12}C$ concentrations (López-Sandoval et al., 2019).

Before analyzing the filters, CRDS-Picarro was calibrated using VPDB standards from the International Atomic Energy Agency (IAEA) including IAEA-CH-6, C3, and 303B with $\delta^{13}C$ values of –10.45 ‰, –24.72 ‰, and +450 ‰, respectively. Additionally, Reston Stable Isotopic Laboratory standards (United States Geological Survey, UGS) were also used, including USG62 (–14.79 ‰), USG40 (–26, 39 ‰), and USG41a (+36.55 ‰, López-Sandoval et al., 2019).



The $^{13}$C and $^{12}$C isotopic mass from the $^{13}$C-enriched sample at the end of incubation time (h) was determined to
calculate PP (µg C$^{-1}$ h$^{-1}$) using the $^{13}$C-CRDS-PP method (López-Sandoval et al., 2019). Particularly, $^{13}$C-PP was
calculated as the isotopic shift of particulate organic carbon (POC) from the samples incubated in the light ($\delta^{13}C_{POC\text{-}Light}$) relative to the dark isotopic composition of the samples ($\delta^{13}C_{POC\text{-}Dark}$). Additionally, the isotopic shift of the
enriched DIC ($\delta^{13}C_{DIC\text{-}Enritched}$) relative to the natural DIC samples ($\delta 13C_{DIC\text{-}Natural}$) was also calculated. The
production was converted to carbon uptake rates considering the particulate organic carbon measured at the end of
incubation in the light-enriched samples per volume filtered in L (POC-µg C L$^{-1}$), and the carbon fixation rate was
calculated per time unit (hours of incubation) following Eq (1) below:

$$^{13}C\text{-}PP = ([(\delta^{13}C_{POC\text{-}Light} - \delta^{13}C_{POC\text{-}Dark})/$$
$$(\delta^{13}C_{DIC\text{-}Enriched} - \delta^{13}C_{DIC\text{-}Natural})]\times POC/v/t)$$

The TB$_{DIC}$ fixation rate (µg C$^{-1}$ h$^{-1}$) was measured using a similar equation as for the $^{13}$C-PP method. However, the
uptake rate was calculated as the isotopic shift of particulate organic carbon from the samples incubated in the dark
($\delta^{13}C_{POC\text{-}Dark}$) relative to the natural isotopic composition of the samples ($\delta^{13}C_{POC\text{-}Natural}$). Additionally, we also
calculated the isotopic shift of the enriched DIC ($\delta^{13}C_{DIC\text{-}Enriched}$) relative to the natural DIC samples ($\delta^{13}C_{DIC\text{-}Natural}$).
The production rate was also converted to carbon uptake rates considering the particulate organic carbon measured
at the end of incubation in the dark-enriched samples and the volume filtered in L (POC-µg C L$^{-1}$). Additionally, the
carbon fixation rate was calculated per time unit (hours of incubation) following Eq (2) below:

$$^{13}C\text{-}TB_{DIC} = ([(\delta^{13}C_{POC\text{-}Dark} - \delta^{13}C_{POC\text{-}Natural})/$$
$$(\delta^{13}C_{DIC\text{-}Enriched} - \delta^{13}C_{DIC\text{-}Natural})]\times POC/v/t$$

The Total$_{DIC}$ fixation was calculated as the sum of the $^{13}$C-PP measured in the light and $^{13}$C-TB$_{DIC}$ measured in the
dark following Eq (3) below:

$$Total_{DIC} \text{ fixation} = {}^{13}C\text{-}PP_{Light} + {}^{13}C\text{-}TB_{DIC\text{-}Dark}$$

The contribution of the TB$_{DIC}$ to the total carbon production was calculated relative to the Total$_{DIC}$ fixation following
Eq (4) below:

$$TB_{DIC}\% = {}^{13}C\text{-}TB_{DIC\text{-}Dark} / Total_{DIC} \text{ fixation}$$

The data for PP and TB$_{DIC}$ were converted to carbon uptake per day using a local photoperiod of 12 hours of
daytime for photosynthesis and 12 hours of nighttime for dark DIC fixation.

### 2.1.5 Bacterial production (BP)



Bacterial production (BP) was measured based on glucose uptake and was expressed in µg C L$^{-1}$ h$^{-1}$ as described by Middelburg et al. (2000). $^{13}$C incorporation was calculated as glucose uptake and defined as the difference between the fraction of $^{13}$C in the natural isotopic composition sample ($F_{Natural}$) relative to the $^{13}$C fraction in the enriched sample ($F_{Enriched}$) following Eq (5) below:

Glucose uptake rate (BP-µg C L$^{-1}$ h$^{-1}$) = [($F_{Enriched}$ - $F_{Natural}$) × POC]/v/t

where F = $^{13}$C / ($^{13}$C + $^{12}$C); which is also expressed as $R/(R+1)$, where R is the carbon isotope ratio obtained from the measured δ $^{13}$C values and it is calculated following Eq (6, Middelburg et al., 2000) below:

$R$ = (δ $^{13}$C/1000 + 1) × V-PDB (Vienna Pee Dee Belemnite),

where V-PDB = 0.0112372. Additionally, the uptake rate was calculated by considering the particulate organic carbon measured in the samples at the end of incubation per time unit (hours of incubation) and per volume filtered in L. The BP rates were reported on a per-day basis considering a 24 hours cycle.

### 2.1.6    Statistical analysis

All statistical analyses were conducted using the JMP PRO 16 software (JMP®, Version <*16.1*> SAS Institute Inc.,
Cary, NC, 1989–2019). A *p*-value ≤ 0.05 was considered statistically significant. The relationship and correlation between variables were explored using Spearman's nonparametric correlations and linear regression, and means were compared using one-way ANOVA and Student's t-test.

### 3    Results

The seawater temperature during the study period ranged from a minimum of 21.0 °C in deep waters (400 m depth) to a maximum of 32.2°C in surface waters, showing no latitudinal changes. Salinity, on the other hand, ranged from 38.21 to 40.80 (Table 1) and increased significantly with latitude (ρ = 0.71, p <0.0001). Our results indicated that
silicate (SiO$_2$) levels were generally low, ranging from 0.44 to 1.65 µM, whereas phosphate (PO$_4$) ranged from undetectable to 0.31 µM. Both SiO$_2$ and PO$_4$ decreased with increasing latitude (ρ = –0.42, p <0.0007, and ρ = – 0.51, p <0.0001, respectively). Nitrate (NO$_3$) (Table 1), however, did not exhibit any significant spatial variability throughout the study, with an average (±SE) low value of 0.57 ± 0.05 µM (Mean ± SE) from the surface to a 30 m depth, and a higher mean value of 2.44 ± 0.55 µM below 30 m depth. Surface (3.66 ± 0.16 m) Chl-a averaged 0.22 ±
0.5 Chl-a µg L$^{-1}$ in the south, 0.38 ± 0.05 µg L$^{-1}$ in the central coastal stations, and 0.34 ± 0.90 µg L$^{-1}$ in the northern stations, and ranged from 0.04 to 0.81 µg L$^{-1}$, with the coastal stations, which exhibited the highest individual values (Table 1). However, during the winter in the northern stations, Chl-a concentration peaked and reached up to 0.69 µg L$^{-1}$, probably reflecting nutrient entrainment due to convective mixing. In addition, Chl-a concentration measured





during CCF cruise through different photic zone layers showed an increase toward the maximum chlorophyll a

depth, located at, on average 61.66 m depth, where it reached up to $0.48 \pm 0.036$ µg L⁻¹.

**Table 1**: Value ranges (minimum-maximum) of the environmental and biological parameters obtained during the study along the Red Sea at different depths. The data in the table are Chlorophyll-a concentration (Chl-a), seawater Temperature (Temp), Salinity (Sal), Nutrient concentrations ($SiO_2$, $PO_4$, and $NO_3$), heterotrophic bacteria abundance

(BACT), and Bacterial Production (BP). N/A: data not available.

| Cruise | Date | Lat °N | Long °E | Depth m | Temp °C | Sal | Chl-a (µg L⁻¹) | $SiO_2$ µM | $PO_4$ µM | $NO_3$ µM | BACT cells mL⁻¹ | BP µg C L⁻¹ d⁻¹ |
|---|---|---|---|---|---|---|---|---|---|---|---|---|
| CCF (Open water) | 16/03/18–21/03/18 | 17.35–22.23 | 38.38–40.42 | 5–90 | 24.20–27.69 | 38.21–38.90 | 0.08–0.69 | 0.57–1.65 | 0.04–0.28 | 0.01–5.31 | N/A | N/A |
| Deep Cruise (Open water) | 04/04/19–09/04/19 | 18.67–24.46 | 37.01–40.22 | 5 | 24.00–27.00 | 38.30–40.00 | 0.14–0.19 | 0.59–1.03 | 0.06–0.20 | 0.14–0.60 | 3.25E+05–7.84E+05 | 0.12 – 0.46 |
| Deep Coral Survey (open water) | 18/01/20–23/01/20 | 22.30–25.75 | 36.34–38.86 | 5 | 23.11–24.90 | 39.62–40.23 | 0.37–0.62 | 0.83–1.03 | 0.01–0.11 | 0.27–1.04 | 4.38E+05–6.57E+05 | 0.04 – 0.08 |
| Time series (coastal) | 21/08/17–05/02/18 | 22.31 | 38.96 | 3 | 24.40–32.10 | 39.30–39.57 | 0.18–0.81 | 0.44–1.02 | 0.01–0.31 | 0.44–1.12 | 2.00E+05–4.13E+05 | |
| Reef (coastal) | 12/11/19 | 22.32 | 39.02 | 3 | 29.90 | 39.78 | 0.40 | N/A | N/A | N/A | N/A | 0.69 |
| Lagoon (coastal) | 22/10/19 | 22.39 | 39.14 | 3 | 32.20 | 40.80 | 0.70 | N/A | N/A | N/A | N/A | 0.92 |
| RSDE (open water) | 07/02/22–05/06/22 | 19.44–24.15 | 36.59–39.44 | 5 | 25.13–28.87 | 38.45–40.16 | 0.07–0.49 | N/A | N/A | N/A | 5.87E+04–1.86E+05 | |
| | | | | 400 | 21.00–21.72 | 39.29–40.54 | N/A | N/A | N/A | N/A | 1.90E+04–3.48E+04 | |

Heterotrophic bacteria cell abundance ranged from $1.90 \times 10^4$ cells mL⁻¹ at 400 m depth to $7.84 \times 10^5$ cells mL⁻¹ in surface waters, averaging $3.78 \pm 0.38 \times 10^5$ cells mL⁻¹ in the photic layer (Table 1). There was no significant difference in bacterial abundance between open and coastal water (F ratio = 0.19, df = 1, p = 0.66). The bacterial production measured as the glucose uptake rate in the dark (HBP) varied from 0.04 µg C L⁻¹ d⁻¹ recorded in the northern stations to 0.69 and 0.92 µg C L⁻¹ d⁻¹ (Table 1) in the coastal reef and lagoon stations, respectively.

Additionally, HBP was significantly different between coastal and open waters (F ratio = 11.07, df = 1, p <0.005), with higher rates observed in the coastal stations ($0.60 \pm 0.21$ µg C L⁻¹ d⁻¹) compared to the open waters ($0.17 \pm 0.04$ µg C L⁻¹ d⁻¹). HBP increased with increasing temperature ($R^2 = 0.71$, p <0.0001).

Primary production rates across the study ranged from 0.84 µg C L⁻¹ d⁻¹ to 47.76 µg C L⁻¹ d⁻¹ (Table 2), and were significantly higher at the coastal stations ($17.77 \pm 3.60$ µg C L⁻¹ d⁻¹) compared to open waters ($7.24 \pm 0.71$ µg C L⁻¹ d⁻¹) (F ratio = 20.17, df = 1, p <0.0001, Fig. 2A). PP also tended to increase with increasing temperature ($\rho = 0.35$, p <0.001), but was independent of Chl-a concentration ($\rho = 0.12$, p = 0.52), and declined with increasing nitrate concentration ($\rho = -0.46$, p <0.014, Fig. 4). Primary production throughout the photic zone (CCF open water)



decreased from 15 µg C L$^{-1}$ d$^{-1}$ at the surface to <1 µg C L$^{-1}$ h$^{-1}$ at the base the photic layer (1 % PAR). In contrast, the concentration of particulate organic carbon across the photic layer changed gradually, from 95.13 ± 10.23 µg C L$^{-1}$ at the surface to 18.25 ± 1.28 µg C L$^{-1}$ at the base of the photic zone.

**Table 2**: Range (minimum-maximum) and mean ± SE of dark DIC uptake rates by bacteria (TB$_{DIC}$), primary
production (PP), the percentage contribution of TB$_{DIC}$ (TB$_{DIC}$ %) relative to the total DIC fixation (Total$_{DIC}$ fixation), and PP obtained at different water depths in different cruises.

| Cruise | Lat (°N) | Long (°E) | Depth (m) | TB$_{DIC}$ (µg C L$^{-1}$ d$^{-1}$) | PP (µg C L$^{-1}$ d$^{-1}$) | PP % | TB$_{DIC}$ % | Mean Depth (m) | Mean PP (µg C L$^{-1}$ d$^{-1}$) | Mean TB$_{DIC}$ (µg C L$^{-1}$ d$^{-1}$) | Mean TB$_{DIC}$ % | Mean PP % |
|---|---|---|---|---|---|---|---|---|---|---|---|---|
| CCF (open water) | 17.35–22.23 | 38.38–40.42 | 5–90 | 1.07–2.69 | 9.84–15.36 | 28.22%–86.82% | 13.18%–71.78% | 3.66 | 11.15 ± 1.67 | 5.23 ± 0.54 | 36 ± 0.02 | 63 ± 0.02 |
| Deep Cruise (open water) | 18.67–24.46 | 37.01–40.22 | 5 | 2.05–12.45 | 2.69–16.71 | 45.48%–70.87% | 29.13%–54.52% | 12.00 | 12.71 ± 0.79 | 2.33 ± 0.11 | 15 ± 0.01 | 84 ± 0.01 |
| Deep Coral Survey (open water) | 22.30–25.75 | 36.34–38.86 | 5 | 2.69–10.61 | 3.16–14.68 | 34.49%–82.68% | 17.32%–65.51% | 28.00 | 10.33 ± 2.04 | 2.04 ± 0.25 | 17 ± 0.01 | 82 ± 0.01 |
| Time series (coastal) | 22.31 | 38.96 | 3 | 3.44–12.69 | 3.46–47.76 | 34.49%–84.37% | 15.63%–65.51% | 45.60 | 5.91 ± 0.68 | 1.89 ± 0.35 | 23 ± 0.02 | 76 ± 0.02 |
| Reef (coastal) | 22.32 | 39.02 | 3 | 3.44 | 4.31 | 55.61% | 44.39% | 61.66 | 2.12 ± 0.69 | 1.66 ± 0.50 | 45 ± 0.04 | 54 ± 0.04 |
| Lagoon (coastal) | 22.39 | 39.14 | 3 | 10.11 | 15.88 | 61.10% | 38.90% | 84.00 | 1.35 ± 0.45 | 1.42 ± 0.35 | 52 ± 0.01 | 47 ± 0.11 |
| RSDE (open water) | 19.44–24.15 | 36.59–39.44 | 5 | 1.89–5.65 | 2.15–6.21 | 48.13%–75.55% | 24.45%–55.43% | 400.00 | N/A | 4.95 ± 1.19 | 40 ± 0.07 | 59 ± 0.07 |
| | | | 400 | 1.37–6.33 | N/A | N/A | N/A | | | | | |


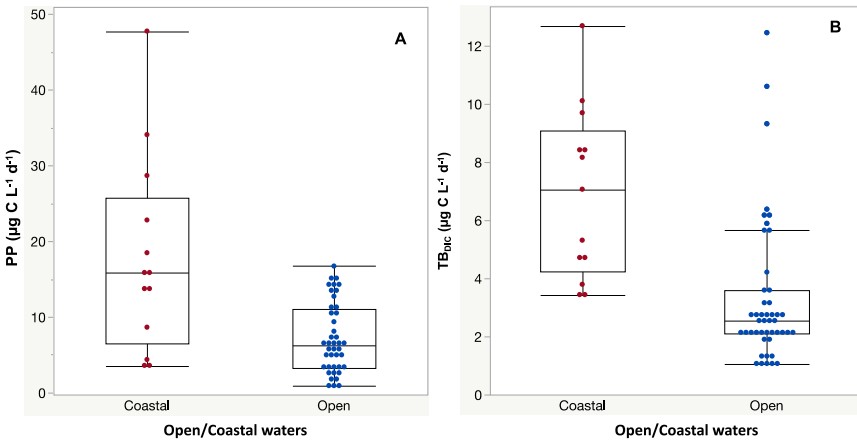

**Figure 2**: (A) Primary Production (PP) and (B) Total dark bacteria DIC fixation (TB$_{DIC}$) measured in open and
coastal waters. Boxplots indicate the 95 % confidence intervals with ±1 SD deviation. The central line in the box represents the median for each group of samples.



The $^{13}$C-dark TB$_{DIC}$ fixation rate varied from 1.07 µg C L$^{-1}$ d$^{-1}$ to a maximum of 12.69 µg C L$^{-1}$ d$^{-1}$ (Table 2) with no latitudinal pattern. We found significant differences in TB$_{DIC}$ between coastal (6.92 ± 0.81 µg C L$^{-1}$ d$^{-1}$) and open

water stations (3.34 ± 0.37 µg C L$^{-1}$ d$^{-1}$); (F ratio = 19.89, df = 1, p <0.0001, Fig. 2B), but not (p = 0.84) between surface (5.24 ± 0.54 µg C L$^{-1}$ d$^{-1}$) and 400 m samples (4.95 ± 1.20 µg C L$^{-1}$ d$^{-1}$) , which showed the highest values compared to other depths (Fig. 3).  Additionally, TB$_{DIC}$ exhibited a weak relationship with HBP, and bacterial abundance (Fig. 4), and on average, TB$_{DIC}$ (4.09 ± 0.38 µg C L$^{-1}$ d$^{-1}$) exceeded the HBP rate (0.26 ± 0.06 µg C L$^{-1}$ d$^{-1}$) by over one order of magnitude. We found no significant correlation between TB$_{DIC}$ and temperature, and nutrient

concentration (Fig. 4). However, TB$_{DIC}$ showed a high significant correlation with POC (ρ = 0.80, p <0.0001), and a weak tendency to increase with increasing Chl-a (ρ = 0.38, p <0.01, Fig. (4).

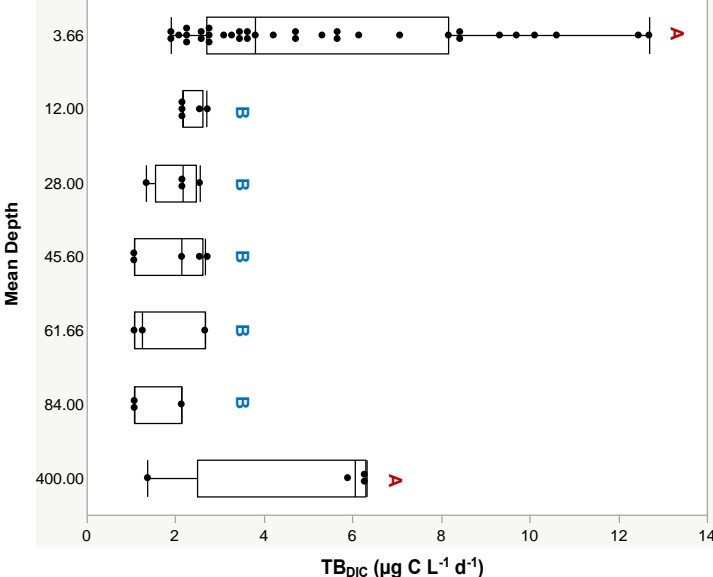


**Figure. 3**: Total bacteria DIC uptake rate (TB$_{DIC}$ µg C L$^{-1}$ d$^{-1}$) measured at the photic zone and in deep waters (400 m), where different letters indicate statistically significant differences ("A" is significantly higher than "B").  Each box plot indicates the 95 % confidence intervals with ±1 SD deviation. The central line in the box represents the median for each group of samples.


The Total$_{DIC}$ fixation rate which is the uptake rate of both TB$_{DIC}$ and PP fixation rate was ranged from 2.03 up to 60.45 µg C L$^{-1}$ d$^{-1}$. We found a significant and positive correlation between TB$_{DIC}$ fixation and PP (ρ = 0.53, p <0.0001, Fig. 4). On average, the contribution of TB$_{DIC}$ % to the Total$_{DIC}$ daily fixation was 33.95 ± 0.02 %, ranging

from 13% to 72% across samples (Table 2). The TB$_{DIC}$ % contribution was independent of the absolute TB$_{DIC}$ fixation rate (R$^2$ = 0.05, p = 0.09), and was negatively correlated with PP (R$^2$ = 0.45, p <0.0001, Fig. 5A). and did not





differ between open and coastal waters (F ratio = 0.008, df = 1, p = 0.92). The high $TB_{DIC}$ at 400 m (1.37 to 6.33 µg C $L^{-1}$ $d^{-1}$, Table 2), implied that the contribution of $TB_{DIC}$ at 400 m was high (40 %) relative to surface PP (Fig. 5B and Table 2).


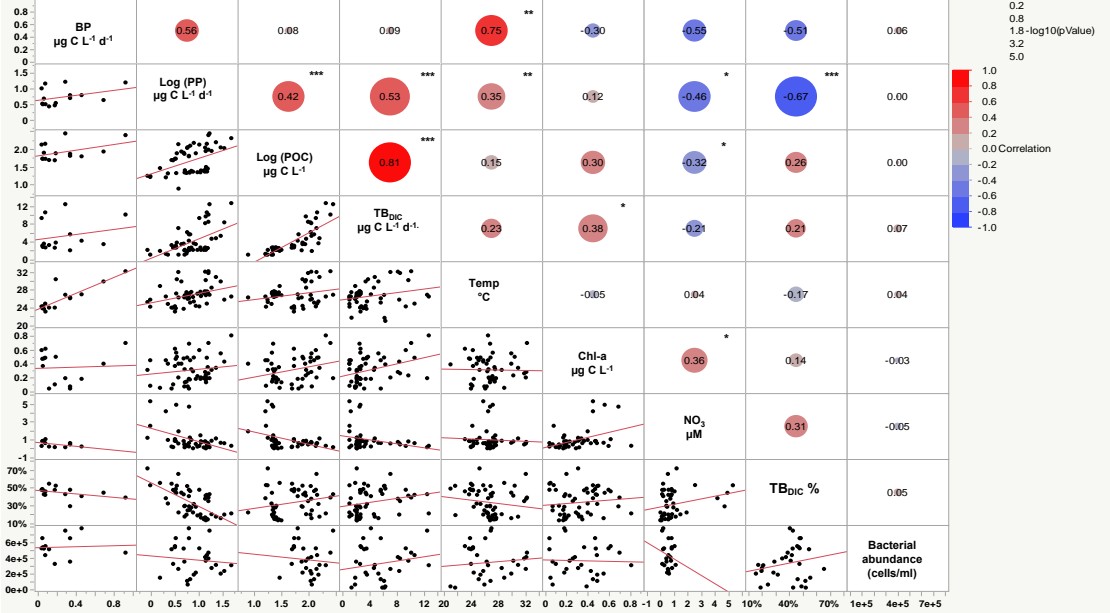

**Figure 4**: Scatterplot matrix plot, lower diagonal, and Spearman correlation coefficient (upper diagonal) among Total Bacterial Dark DIC uptake ($TB_{DIC}$), Total Bacterial Dark DIC uptake contribution ($TB_{DIC}$ %), Primary

Production (PP), Particulate Organic Carbon (POC), Temperature (Temp), Bacterial Production (BP), Nitrate ($NO_3$), Chlorophyll -a concentration (Chl-a), and bacterial abundance. * p <0.01, ** p < 0.001, *** p <0.0001.

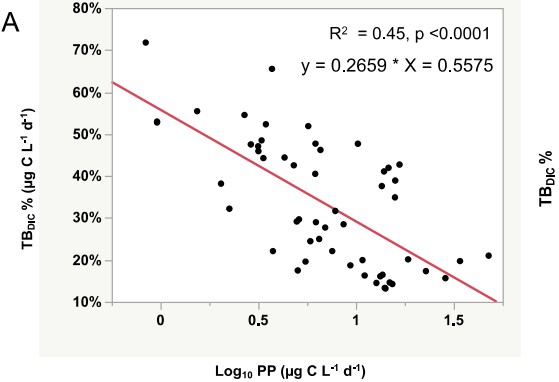

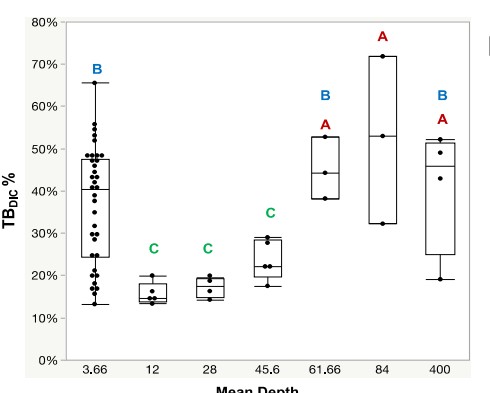





**Figure 5**: Figure (A) shows the significant probability of the relationship between the percentage contribution of
TB$_{DIC}$ (TB$_{DIC}$ %) and the primary productivity (PP), whereas (B) shows the total bacterial DIC uptake contribution
(TB$_{DIC}$ %) to the total DIC fixation within the photic zone and the deep 400 m. The boxes with the same letters are
not statistically significant, as determined by pairwise comparisons via Student's t-test.

## 4    Discussion

Although the contribution of heterotrophic bacteria to dark $CO_2$ fixation was discovered more than 80 years ago
(Wood & Werkman, 1936), quantitative assessments of dark heterotrophic bacteria $CO_2$ fixation remain few (Braun
et al., 2021), particularly relative to primary production rates. Our study successfully used the light and dark [13]C-
bicarbonate additions coupled with CRDS-Picarro to accurately quantify the DIC incorporation by phytoplankton,
heterotrophic bacteria, and chemo-autotrophs during daytime and nighttime, respectively. Decades after S. Nielsen
(1952) introduced the [14]C method to measure phytoplankton primary production, the [13]C-PP method has recently
garnered increasing attention as an alternative to the use of radioactive isotopes.  Here, we extend the use of [13]C-
bicarbonate additions to also resolve dark DIC uptake by bacteria, providing, to the best of our knowledge, the first
application of this method coupled with CRDS-Picarro. Indeed, most previous studies measured dark DIC fixation
using the [14]C-bicarbonate method (Baltar et al., 2010; Reinthaler et al., 2010; Yakimov et al., 2014; Zhou et al.,
2017; Alonso-Sáez et al., 2010; Signori et al., 2017; Llirós et al., 2011) or genome and 16S ribosomal analysis
(González et al., 2008; Yakimov et al., 2014), whereas very few studies have used [13]C-bicarbonate to measure dark
DIC fixation (Roslev et al., 2004). One of the main advantages of using the stable isotope instead of radioactive
isotopes additions is that it greatly reduces the health and safety issues associated with using radioactive products.
Additionally, this method can be easily applied to characterize field samples (Middelburg et al., 2000; Boschker and
Middelburg, 2002). Recent studies have demonstrated that the phytoplankton photosynthesis quantification results
obtained with the 13C-PP method coupled with CRDS-Picarro were similar to those achieved with the 14C-method
(López-Sandoval et al. 2018; López-Sandoval et al. 2019). Cavity ring down spectroscopy (CRDS), particularly
with the Picarro analyzer, offers several advantages for stable isotope analysis applications. CRDS can detect trace
amounts of isotopes in samples with very high sensitivity. This makes it particularly useful for analyzing low-
concentration samples like oligotrophic waters. The high sensitivity of CRDS also allows for high precision and
accuracy in isotopic measurements. (Berden et al., 2000; López-Sandoval et al., 2019).

Previous studies have confirmed that bacterial dark $CO_2$ uptake contributes significantly to the carbon flux dynamics
of oligotrophic waters, shallow waters, coastal waters, and deep seas (Alonso-Sáez et al., 2010, Yakimov et al.,
2014, Zhou et al., 2017., Signori et al., 2017, and LLiro's et al., 2011). Here, we confirmed the relevance of the dark
$CO_2$ fixation processes in the oligotrophic Red Sea. Our estimates were within the range of reported dark
DIC fixation rate, which range from 0.001 µg C L$^{-1}$ d$^{-1}$ in surface waters of the Subtropical North Atlantic and
tropical estuarine systems (Reinthaler et al., 2010; Signori et al., 2017) to 206 µg C L$^{-1}$ d$^{-1}$ in an eutrophic
Mediterranean coastal lagoon (LLiro's et al., 2011). In contrast, the highest values recorded in our study were found

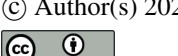



in coastal waters and were similar to those reported from oligotrophic ocean waters (Alonso-Sáez et al., 2010;
LLiro's et al., 2011; Zhou et al., 2017). The deepest dark $CO_2$ fixation rate measurement yet reported was carried
out in the Mediterranean Sea at 4900 m, estimated at $0.096 \pm 0.02$ (Yakimov et al., 2014). We recorded a relatively
high $TB_{DIC}$ uptake rate in Red Sea surface waters, toward the base of the photic zone but reaching high values at 400
m. Increases in dark DIC fixation rates at depth have been observed in the tropical South China Sea at depths

between 200 and 1500 m, with rates even exceeding those at the surface (Zhou et al., 2017). The high $TB_{DIC}$ values
reported in the surface and deep water in our study suggest that dark DIC fixation not only contributes significantly
to the carbon fixation dynamics of surface water, where bacterial abundance is high, but also throughout the entire
(oxygenated) water column (Reinthaler et al., 2010; Yakimov et al., 2014; Zhou et al., 2017). Therefore, given the
high $TB_{DIC}$ values measured in the oligotrophic Red Sea ecosystem, our findings highlighted the importance of

accounting for dark DIC fixation in total carbon production estimations.

The Red Sea is characterized by a warm temperature throughout the water column (Shaltout, 2019), with average
temperature of $21.46 \pm 0.23$ °C at 400 m depth recorded in our study. Here, temperature was found to have a
positive correlation with PP but no correlation with the $TB_{DIC}$ uptake rate, whereas Chl-a had no correlation with PP
as PP decreased gradually down the photic zone while Chl-a showed the maximum values in the bottom of the

photic zone. However, Chl-a showed a positive correlation with $TB_{DIC}$. Nitrate showed a negative correlation with
PP, suggesting nitrate depletion by the more productive phytoplankton communities, whereas $TB_{DIC}$ was positively
correlated with PP. Studies conducted in the North Atlantic Ocean and tropical estuarine systems have reported a
weak relationship between dark $CO_2$ fixation and temperature, nutrients, and Chl-a (Reinthaler et al., 2010; Signori
et al., 2017). In contrast, a study conducted in the South China Sea reported that dark $CO_2$ fixation rates increased in

with temperature and nutrient concentration (Zhou et al., 2017).

The significant negative relationship between $TB_{DIC}$ % and PP confirmed the relevance of the dark $CO_2$ fixation
toward the most oligotrophic waters. Bacterial dark $CO_2$ fixation rate ($TB_{DIC}$) contributed significantly to the total
DIC uptake within the photic zone, where it reached up to 72% of the photosynthetic primary production and up to
52% of the surface PP in the deep water at 400 m. In the eastern North Atlantic Ocean, dark $CO_2$ fixation was

reported to support 72% of the prokaryotic carbon demand in the mesopelagic layers below 200 m (Baltar et al.,
2010). Baltar and Herndl (2019) analyzed data collected over the course of 30 years and found that the dark
$CO_2$ uptake rate contributed up to 22% of the total PP at the euphotic layer (0–150 m). Additionally, increasing
evidence has suggested that dark $CO_2$ uptake by heterotrophic bacteria contributes significantly to surface
$CO_2$ fixation, contributing up to 30% of the DIC uptake in some oligotrophic waters (González et al., 2008;

Palovaara et al., 2014). Similarly, in our study we recorded an average contribution of $TB_{DIC}$ to total DIC uptake of
33.95 % within the photic zone and the deep 400 m water. The increase in the contribution of $TB_{DIC}$ to the carbon
flux with decreasing PP in the Red Sea highlights the importance of dark DIC fixation as a key mechanism driving
plankton communities, particularly in highly oligotrophic environments with low (surface) or absent (deep water)
primary production. Additionally, the relevance of dark $CO_2$ incorporation is likely significant in oligotrophic and

nutrient-depleted environments where the availability of labile organic carbon is limited (González et al., 2008,



Alonso-Sáez et al., 2010). Overall, our results confirm the relevance of dark $CO_2$ fixation to the oligotrophic Red Sea.

Our findings indicate that $TB_{DIC}$ exceeded HBP, as reported in previous studies (Zhou et al., 2017), confirming the important role of $TB_{DIC}$ in fueling bacterial metabolism in oligotrophic waters, with low levels of labile organic
matter for bacterial growth such as surface and deep oligotrophic waters. The high $TB_{DIC}$ contribution to total DIC uptake can support the growth of the bacterial community, providing a path to support bacterial metabolism, respiration and carbon flux in the microbial loop (Zhou et al., 2017). The importance of dark $CO_2$ fixation processes seemed to increase with depth in the Red Sea, as the dark/light ratio of $CO_2$ fixation rate increased in deeper waters, reaching up to $1.13 \pm 0.65$ toward the base of the photic zone. These findings were consistent with those of Baltar
and Herndl (2019), who reported that the dark/light ratio reached 1 at 120–160 m depths from data collected along the ALOHA and BATS; the longest oceanic time series of the Atlantic and Pacific Ocean, highlighting the urgent need to account for dark DIC fixation in future studies on total primary production. Considering the total net primary production in the ocean to be approximately 50 Pg C $y^{-1}$ as reported by Field et al. (1998) and based on the potential $TB_{DIC}$ % contribution of 13.18% to 71.78% to the $Total_{DIC}$ fixation reported in our study, we estimated that
approximately 6.5 to as much as 35.5 Pg C $y^{-1}$ could be added to the global primary production estimation. This would be a considerable amount of carbon productivity by heterotrophs that is not being accounted for in current carbon flux and production estimations, which could represent a significant source of carbon in surface and deep waters (Baltar and Handle, 2019).

### 5 Conclusions

Using a stable isotope method, our study demonstrated the substantial contribution of dark $CO_2$ assimilation by heterotrophic bacteria in the oligotrophic Red Sea. The results presented herein represent a first attempt to estimate and confirm the role of dark heterotrophic bacteria $CO_2$ assimilation to the carbon flux dynamics along the Red Sea at different depths and in different water bodies. Even though temperature, a uniquely influential feature of the warm
Red Sea, appeared to have a weak correlation with $TB_{DIC}$, our study confirmed that the importance of anaplerotic $CO_2$ incorporation and chemo-autotrophic process would be significant in environments with low or absent primary productivity. Moreover, due to the large fraction of $Total_{DIC}$ fixation generated from the contribution of $TB_{DIC}$ in the surface and the deep water, as reported in our study and other studies, it is essential to account for the contribution of heterotrophic dark $CO_2$ fixation to the total DIC fixation as a source of prokaryotic carbon demand.

*Data availability.* The data are presented in the manuscript; and can be requested from the corresponding author

*Author contribution.* Each of the authors made substantial contributions to the conception, design, and execution of this study. Prof. Susana Agustí was responsible for the development of the study design and goals, data analysis,
critical revisions of the manuscript, and overall project coordination. Prof. Carlos Duarte contributed to the development of the study design, interpretation of the data, and critical revisions of the manuscript, in addition to his



coordination to conduct the project. In addition, Prof. Carlos provided the raw data run by CRDS-Picarro in his laboratory. Dr. Daffne López-Sandoval was involved in the data collection and analysis, and provided critical feedback on the manuscript. Afrah Alothman was responsible for data collection, data analysis and drafting the

manuscript. In addition, Afrah reviewed any critical revisions made by the co-authors. All authors have read and approved the final manuscript for submission.

*Competing interests*. The authors declare that they have no conflict of interest.

*Acknowledgment:* We would like to express our deepest gratitude to each and every one involved in this study, and the King Abdullah University of Science and Technology who have made this research possible. First and foremost, we would like to thank and show our grateful to all participants who generously shared their time and experiences to enclosed this research including all the crew members and the scientific leaders of the vessels and cruises involved in this study. In addition, we would like to thank Coastal & Marine Resources Core Lab (CMR) team at King

Abdullah University of Science and Technology (KAUST) who were behind the possibility of conducting coastal experiments outdoor set up. We also extend our appreciation to Mongi Ennasri who provided the support and assistance throughout the samples analysis in the CRDS-Picarro. Each and every one of these individuals and organizations support enabled us to collect and analyze the data, and to disseminate our findings.

*Financial support.* This work was supported by King Abdullah University of Science and Technology (KAUST).

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
