# Peer review of "Bacterioplankton dark CO2 fixation in oligotrophic waters"

_Biogeosciences, 2023_

## Author Response (AR1)

**Reviewer 1# Revision of "Bacterioplankton dark CO2 fixation in oligotrophic waters" by Alothman et al.**

**General comments**

This is an exciting paper dealing with a very topical issue, which is the contribution of dark CO2 fixation to primary production in the ocean, in particular in oligotrophic waters. Overall the manuscript is well structured and the research strategy is sound, so it should be a valuable contribution to the readerships of Biogesciences. I have included some detailed comments below that I would like the authors to address — some are trivial, some probably arise from my own misunderstandings, and some should be addressed before publication.

**Authors**: We thank the reviewer for the useful comments, we detail below how we addressed the concerns and suggestions in the revised manuscript.

**Specific comments (in order of appearance)**

Reviewer #1 line 37. Bracket missing in Wood and Werkman (1936

Authors: Bracket has been added.

Reviewer #1 lines 46-51. Summarize and merged these 2 sentences into 1 to avoid repetition.

Authors: The two sentences in the lines 46-51 have been combined as suggested.

Reviewer #1 line 53. I cannot seem to have access to the original Romachenko 1964 paper and I don't remember; but can you please confirm it was estimated to contribute up to 8 % of heterotroph carbon biomass production in the ocean? I mean, that the study was done with oceanic/marine waters?

Authors: This publication has been removed and the study has been replaced by a smiler one cited as follows: "In general, anaplerotic CO2 fixed by some bacterial species has been reported to increase bacterial cell production by 1-6.5 % in incubated cultures (Roslev et al., 2004)"

Reviewer #1 lines 57-60. This sentence "Genetic studies have demonstrated that increases in the abundance of associated anaplerotic enzyme transcripts coincided with a sudden increase in bacterial metabolism, which could contribute significantly to the total DIC fixation rates in oligotrophic environments (Baltar et al., 2016)", is partly true but a small detail is missing to make it more complete, which is that the increase in anaplerotic activity was found in response to the addition of organic matter. So, yes, the experiment was performed in an oligotrophic environment (i.e. open deep ocean), but in response to the addition of organic matter. Maybe good to include it in there to make the argument more precise.

Authors: The sentence has been corrected as follows: "Additionally, a genetic study conducted in the Atlantic Ocean has demonstrated that organic matter enriched samples showed an

increase in the abundance of associated anaplerotic enzyme transcripts coincident with a sudden increase in bacterial abundance (Baltar et al., 2016)".

Reviewer #1 lines 60-62. The sentence "Additionally, the recent discovery of light-driven CO2 incorporation by proteorhodopsin-containing flavobacterium Polaribacter sp. highlighted the significant role of anaplerotic metabolism in heterotrophs (González et al., 2008)." This study in not so recent is from 2008, so I would remove "recent". Also, to follow a logical chronological order I would write this sentence before the previos one (actual l.57-60).

Authors: The sentence has been moved before the previous sentence and corrected as follows: "A discovery of light-driven CO2 incorporation by proteorhodopsin-containing flavobacterium Polaribacter sp highlighted the significant role of anaplerotic metabolism in heterotrophs life (González et al., 2008).

Reviewer #1 lines 62-65. I would mention briefly in that sentence what is the range of contribution suggested in that cited study.

Authors: The range has been added and the sentence has modified as follows: "Therefore, whereas the total primary production of oceanic ecosystems is typically attributed to photosynthesis, dark chemo-autotrophic and anaplerotic metabolism can also contribute from 5-22 % to the total DIC fixation (Baltar and Herndl, 2019).

Reviewer #1 lines 68. In the sentence "These features result in a limited nutrient input and an oligotrophication gradient from south to north", please specify if is increasing or decreasing from S to N, so that the readers can follow the argument clearly.

Authors: The sentence has been specified and modified as follows: "These features result in a limited nutrient input and an oligotrophication decreasing gradient from south to north (Wafar et al., 2016).

Reviewer #1 lines 73-79. Would be good idea to include hypothesis/es in this last paragraph of the introduction.

Authors: A hypothesis has been included in the last paragraph of the introduction as follows: "In our study, we hypothesized that there is a substantial involvement of dark CO2 fixation to the total DIC fixation within the Red Sea, increasing toward the most oligotrophic waters.

Reviewer #1 lines 85-95. I would already cite here Table 1.

Authors: The table has been cited in the first paragraph of the method as suggested.

Reviewer #1 lines 122-125. This was not clear to me: Why were the samples from the CCF cruise filtered by a GF/F and the others by 3 µm? was this filtration with GF/F done before the incubation? If that is right then probably you missed the fraction of ca. 0.7-3 µm in the experiment from the CCF cruise? Please rewrite for clarity.

Authors: We corrected the sentence in the revised manuscript to avoid misunderstandings: "After incubation, dark and light bottles samples from CCF cruise were filtered through pre-combusted Whatman GF/F filters (López-Sandoval et al., 2021). Incubated samples from other cruises were first filtered through 25 mm diameter 3 µm Silver membranes (STERLITECH) followed by a filtration through 0.2 µm pore size silver membrane filters (25 mm diameter, STERLIECH). The collected filters were placed in small Petri dishes containing 150 µl (50 %) HCl to remove carbonate from the filters, allowed to dry for 12 hours, and stored at −20 °C until required for downstream analyses. Moreover, the natural isotopic composition of particulate organic carbon was measured in the surface and deep seawater at each station in similar filters as indicated above (Whatman GF/F filters in CCF cruise and 0.2 µm pore size silver membrane the rest of cruises)."

Reviewer #1 lines 122-125. I would merge these two sentences, by saying that samples were filtered first by 3 µm followed by 0.2 µm.

Authors: The two sentences have been merged as shown above.

Reviewer #1 lines 227-228. Was it expected to not find a latitudinal gradient in temperature? Maybe because samplings were done at different times of the year? Might be good to mention.

Authors: All data were collected during different time of the year; therefore, no latitudinal gradients were found.

The sentence has been modified as follows: "The seawater temperature during the study period ranged from a minimum of 21.0 °C in deep waters (400 m depth) to a maximum of 32.2°C in surface waters. As data were collected during different seasons, surface temperature didn't show a clear latitudinal pattern"

Reviewer #1 lines 234-238. The sentence "Surface (3.66 $\pm$ 0.16 m) Chl-a averaged 0.22 $\pm$ 0.5 Chl-a µg L-1 in the south, 0.38 $\pm$ 0.05 µg L-1 in the central coastal stations, and 0.34 $\pm$ 0.90 µg L-1 in the northern stations, and ranged from 0.04 to 0.81 µg L-1, with the coastal stations, which exhibited the highest individual values (Table 1)" is difficult to follow. Consider splitting it up. Also sounds extrange that the average is 3.66, where as the averages of all others (central coastal, northern,...) are at least one order of magnitude higher, but maybe is because I misunderstood this sentence.

Authors: We modified the sentence in the revised manuscript. Indeed, 3.66 $\pm$ 0.16 m was the averaged surface depth, that we now removed to increase clarity.

The sentence now reads: "Surface phytoplankton Chl-a concentration averaged 0.22 $\pm$ 0.5 µg L-1 and 0.34 $\pm$ 0.90 µg L-1 in the south and north stations, respectively. Coastal stations showed a close averaged Chl-a value of 0.38 $\pm$ 0.05 µg L-1, although exhibited highest individual values (Table 1)".

Reviewer #1. Table 1. In the caption it is mentione that N/A is used for not available data, but there are still empty cells with nothing written in the Table.

Authors: The table has been corrected and N/A was added to the empty cells

Reviewer #1. Table 2. I could not follow the depth description. The 4th column shows the "Depth (m)", and then the 9th column the "Mean Depth (m)", but they do not seem to match. E.g. the Deep Cruise had a Depth of 5m but a Mean Depth of 12 m. Please revise the depths of all the cruises in this table. Also, the last row seems to be misplaced. And no need to add decimals in the Mean Depth numbers.

Authors:  We agree the fourth column, representing the averaged depth, was confusing and unnecessary as this data is reported in Figure 3; we removed the mean depth data in the revised version of the manuscript.

Reviewer #1. Figure 3. Also no need of 2 decimals in the depth numbers (y-axis).

Authors:  Figure 3 has been corrected and decimals removed.

Reviewer #1. Figure 4. Consider adding units to the X- and Y-axis.

Authors:  Units are included in the parameter's tittles.

Reviewer #1. In the y-axis there is the TBDIC (TBDIC %) which should have % as units, but there are other units (of concentration) included as well. Please revise.

Authors: The figure has been corrected.

Reviewer #1 Figure 4 and 5. I am not an expert on the calculations of isotopic signatures, but please double check that there is not autocorrelation in Figure 5 and in some of the relations (the ones derived from isotopic signatures) in Figure 4. What I mean is to check for example that the relation described in Fig.5A is not just an artifact because the parameter in X-axis or in the Y-axis were calculated including some common variable/s. I checked in the methods section how the TBDIC % and the PP were calculated, and it looks like there might be some common variables, but cannot say for certain. I would double-check all the correlations for autocorrelation, and if there is not autocorrelation I would include a sentence somewhere in the text saying/explaining this point.

Authors: PP and TBDIC were calculated from different filters using different incubations. For PP there was a separate dark and light incubations, while for TBDIC, it was calculated from a separate dark incubation related to the initial, so there is not autocorrelation when it comes to isotopic signatures. The TBDIC % is calculated from the sum of the both TBDIC and PP DIC

fixation relative to the TBDIC fixation, and the linear regression was built to show the relationship of the variables' ratio with respect to each other to identify positive or negative trends, and was only significantly and negatively correlated with PP.

We added the following sentence (lines 351-352, 376-377) to the revised manuscript: "Here, we independently calculated dark DIC uptake and PP, and we confirmed the relevance of the dark CO2 fixation processes in the oligotrophic Red Sea", "TBDIC % contribution was calculated from TBDIC uptake rate relative to the sum of the independent PP measured in the light and the independent TBDIC measured in the dark".

Reviewer #1, lines 326-328. I would rephrase this sentence" quantitative assessments of dark heterotrophic bacteria CO2 fixation remain few". I might be mistaken with saying that the contribution of CO2 fixation is low, when I think what you refer to is that what is low is the number of studies on this topic.

Authors: The sentence has been corrected to: "Although the contribution of heterotrophic bacteria to dark CO2 fixation was discovered more than 80 years ago (Wood & Werkman, 1936), the number of studies quantifying dark heterotrophic bacteria CO2 fixation is scarce"

Reviewer #1 line 348. Remove the period (.) before the citations.

Authors: period has been removed.

Reviewer #1 line 359-360. I would remove one "at depth", since you mention it 2 times in the same sentence.

 Answer: The repeated word "at depth" has been removed.

Reviewer #1 lines 367-368, 371-372, 376-377. Depending on the autocorrelation point I mentioned before these sentences might need to be modified, and their associated further statements.

Authors:  As indicated above, the following sentence has been added to the revised manuscript to clarify this aspect: "TBDIC % contribution was calculated from TBDIC uptake rate relative to the sum of the independent PP measured in the light and the independent TBDIC measured in the dark".

Reviewer #1 line 427. Instead of Prof. Carlos better write also de surname.

Authors: The surname has been written instead of first name.

I would like to stress that this is a very good paper, and will be better once the wrinkles are ironed out.

**Reviewer 2# Revision of "Bacterioplankton dark CO2 fixation in oligotrophic waters" by Alothman et al.**

**Detailed comments to the author**

The manuscript (MS) of Alothman and co-workers reports on Carbon fixation data in oligotrophic warm waters. The authors performed an approach based on the use of 13C instead of 14C incorporation rates. Data and results are of interest of BGD audience and the whole scientific community either dealing with C fixation rates, global C cycle and biogeochemical processes mediated by prokaryotes.

The MS in its current form is suitable for publication if some minor modifications are addressed (see below).

Authors: We thank the reviewer for the useful comments, we detail below how we addressed the concerns and suggestions in the revised manuscript.

Reviewer # 2. On the Methods section it would be great if authors provide with information (either by a table or in the body of the MS) on water samples in each sampling site. It's not clearly reflected in the present MS and a bit confusing when refered to the previous work conducted by the main part of the list of authors.

Authors: We revised the description of the water sampling adding more clear information. In lines 89- 96 of the revised manuscript: "Water samples during this study were collected from surface water ranging from 3-5 m (CCF, DC, DCS, RSDE, Time series and coastal stations), adding four to five different photic layers depths in the water column, ranging from 12-90m during the CCF cruise), and deep-water samples collected at 400 m during the RSDE cruises. Water samples were collected in the early morning using 12 L Niskin bottles with a rosette system (López -Sandoval et al., 2021) or 10 L Niskin bottles deployed manually for some surface water sampling. For the deep water (400 m), the water samples were collected using 1.5 L Niskin bottles attached to a remotely-operated underwater vehicle (ROV) or in a submarine for RSDE on board R/V OceanXplorer. Data of seawater temperature, salinity, and underwater photosynthetic active radiation were obtained from CTD deployments for the studied coastal and open water stations (Table 1) as described in López-Sandoval et al. (2021)."

Reviewer # 2. I would like to add just a final general comment with respect to graphs and figures which appears to be small (e.g., figure 2 and 5 with respect to figure 3, or figure 4 itself).

Authors: The figures have been modified to homogenize the sizes.

**Point by point comments**

Reviewer # 2 line 13. Authors refers to heterotrophic bacterial production and bacterial production in the MS, however there is no way to discriminate between bacterial and archaeal production. Accordingly, maybe it's more proper refer to prokaryotic production.

Authors: We agree and we included a paragraph indicating the relevance of archaea contribution to the heterotrophic bacterial production and dark DIC uptake in the revised manuscript. We did not extend to "prokaryotic" to avoid including other prokaryotes as the cyanobacteria Prochlorococcus and Synechococcus, which are present in the Red Sea waters and then it could contribute to complicate the concepts.

In lines 399-404 as follows: "Although in our study we refer to bacterial production, archaea can contribute to BP and in a higher proportion to the dark DIC uptake than bacteria. A study conducted in the northern Red Sea (Gulf of Aqaba) indicated that 10–15 % of leucine incorporation could be attributed to archaeal activity or to bacteria unaffected by the added antibiotics (Ionescu et al., 2009). However, they found that the impact of adding antibiotics to the incubations largely inhibited dark $CO_2$ incorporation, although it was minor below the photic layer, suggesting that a large proportion of the dark DIC uptake was due to archaea in the deep waters (Ionescu et al., 2009).

Reviewer # 2 line 37. please consider adding a final bracket ")" just after "(1936".

Authors: bracket has been added.

Reviewer # 2 line 54. please consider replacing "Lliro´s" by "Llirós" here and all through the text.

Authors: Lliro's has been replaced by Llirós through the text.

Reviewer # 2 line 103. please consider replacing "analysis" by "analyses" here and all through the text where more than one analysis was made.

Authors: The word analysis has been changed to analyses.

Reviewer # 2 line 121. please consider add something like "in concordance with CTD profiles" just after "at the assigned depth" to clarify.

Authors: The sentence has been modified accordingly as follows: "The bottles for PP were covered with neutral-density nets to reduce the light intensity according to the matching light received at the assigned depth in concordance with CTD profiles"

Reviewer # 2 line 135. please consider removing "The" just after "water".

Authors: "The" has been removed.

Reviewer # 2 line 137. please consider using plural of "pigment" and the corresponding verb.

Authors: The noun and the verb have been corrected.

Reviewer # 2 line 138. please consider using italics for "a" in "Chl-a".

Authors: italic has been used for a in Chl-a in all the text

Reviewer # 2 line 178-179. is the formula correct calling "v" after POC?

Authors: "v" in the formula is now added to the text to explain that v after POC is used to calculate the production per liter, and v presented the sample volume incubated and filtered.

Reviewer # 2 line 199. do the TBdic% formula needs the addition of "x100"?

Authors: Yes, that's right. It is now added to the formula.

Reviewer # 2 line 302. please consider removing "was" after "ranged".

Authors It has been removed

Reviewer # 2 line 327. please consider replacing "remain few" by "are scarce".

Authors: it has been changed to "are scarce"

Reviewer # 2 line 329. please consider removing "the" before "DIC".

Authors: "The" has been removed

Reviewer # 2 line 335. please consider re-oder the citation list.

Authors: The citation list has been re-ordered.

Reviewer # 2 line 339. please consider removing extra "s" in "additions".

Authors: "s" has been removed

Reviewer # 2 line 354. please reconsider classifying Coromina lagoon as a coastal one taking into account that there is rougly 50 Km distance in straight line between the lagoon and the Mediterranean Sea.

Authors: word "coastal" has been removed.

Reviewer # 2 line 357. please provide with units for dark CO2 fixation for Yakimov data.

Authors: Unit has been added.

Reviewer # 2 line 401. please consider adding "datasets" just after "BATS".

Authors: "datasets" word has been added

Reviewer # 2 line 406. please consider replacing "heterotrophs" by "heterotrophic microbes".

Authors: "heterotrophs" has been replaced by "heterotrophic microbes".

Reviewer # 2 Figure 1. please provide with a better contrasted image taking into account colour codification and visibility (blue, green and dark dots or country text are not clearly seen). Please consider helpful tools as Colorbrewer (https://colorbrewer2.org/#type=qualitative&scheme=Set1&n=9)

Authors: The dots and the text color have been fixed.

Reviewer # 2 Figure 2. please consider invert the order of "open" and "coastal" terms in figure caption or in figure axis caption.

Authors: The order has been corrected in the axis caption

Reviewer # 2 Figure 3. please provide with depth range for the measure photic zone in the figure caption. Furthermore, are the plotted graphs mean depths for the distinct cruises (values reported in table 1 and 2?).

Authors: The range depth has been provided. The plotted graphs for the all the data presented in the table 2 including different cruises together.

Reviewer # 2 Figure 4. please consider replacing comas (",") before and after "lower diagonal" by parenthesis ("(").

Authors:  The "," has been replaced by "()".

Reviewer # 2 Figure 5. please consider inverting the graphs axis, then depth will be in the 0Y axis, and also consider scaling TBdic to 100%.

Authors: The figure has been corrected accordingly.

Reviewer # 2 Table 1. please us "N/A" or provide "0" value for empty cells. Furthermore, depth values referes to ranges for CCF samples?

Authors: N/A has been provided for all empty cells. The depth is ranged for CCF cruise (different photic layer depths).

Reviewer # 2 Table 2. please provide with a better way of presenting range and minimum-maximum values described in Table 2. Why there are not mean values for 5 m in RSDE samples?

Authors: The table has been modified for better understanding and to avoid confusion only ranged data presented here, while mean data presented in the different figures.